# Interstitial Cells of Cajal—Origin, Distribution and Relationship with Gastrointestinal Tumors

**DOI:** 10.3390/medicina59010063

**Published:** 2022-12-28

**Authors:** Petru Radu, Mihai Zurzu, Vlad Paic, Mircea Bratucu, Dragos Garofil, Anca Tigora, Valentin Georgescu, Virgiliu Prunoiu, Florian Popa, Valeriu Surlin, Victor Strambu

**Affiliations:** 1General Surgery Department, Carol Davila Nephrology Hospital Bucharest, 020021 Bucharest, Romania; 2Oncological Institute “Prof. Dr. Alexandru Trestioreanu”, 022328 Bucharest, Romania; 3Sixth Department of Surgery, University of Medicine and Pharmacy of Craiova, Craiova Emergency Clinical Hospital, 200642 Craiova, Romania

**Keywords:** interstitial cells of Cajal, cancer, gastrointestinal tumors, c-kit

## Abstract

The interstitial cells of Cajal (ICC) represent a particular network formed by some peculiar cells that were first described by the great neuroanatomist, S. Ramon y Cajal. Nowadays, the ICC have become a fascinating topic for scientists, arousing their curiosity; as a result, there is a vast number of published articles related to the ICC. Today, everybody widely accepts that the ICC represent the pacemaker of the gastrointestinal tract and are highly probable to be the origin cells for gastrointestinal tumors (GISTs). Recently, Cajal-like cells (ICLC) were described, which are found in different organs but with an as yet unknown physiological role that needs further study. New information regarding intestinal development indicates that the ICC (fibroblast-like and muscle-like) and intestinal muscle cells have the same common embryonic cells, thereby presenting the same cellular ultrastructure. Nowadays, there is a vast quantity of information that proves the connection of the ICC and GISTs. Both of them are known to present c-kit expression and the same ultrastructural cell features, which includes minimal myoid differentiation that is noticed in GISTs, therefore, supporting the hypothesis that GISTs are ICC-related tumors. In this review, we have tried to highlight the origin and distribution of Cajal interstitial cells based on their ultrastructural features as well as their relationship with gastrointestinal stromal tumors.

## 1. Introduction

The story of the ICC is a fascinating story of ever-changing medical concepts that emerged as a result of the interplay between researchers’ limitless scientific intuition and its formal constraints through the continuous development of medical methodology but always limited [1]. It is this interplay that lies behind the evolution of fundamentally correct concepts many decades before the methods moved to the stage of providing sound theories if not providing the evidence itself [1].

Thus, the great Spanish neuroanatomist, Santiago Ramon y Cajal, conducting studies on these cells, arrived at the hypothesis (1893, 1911) that networks of interstitial cells anastomosed to each other are influenced “primarily” by components of the nervous system, while interstitial cells (seen as primitive “accessory” neurons) exert direct regulatory effects on the contraction of smooth muscle in the gastrointestinal tract [1].

Despite the fact that more than a century has passed since Ramon y Cajal described the staining characteristics of the “interstitial nerve cell” located between the external longitudinal muscle and the circular muscle of the intestine at the level of the Auerbach’s plexus (1893, 1911), the function and developmental origin of these cells have remained unclear [2]. Although Ramon y Cajal and other contemporary researchers believed that these interstitial cells of Cajal were primitive neurons, it has been suggested that these cells are specialized smooth muscle cells [3], while other researchers have characterized them as fibroblasts [4].

Independent of Cajal’s research, Sir Arthur Keith (1914, 1915), the scientist who described the sino-atrial cardiac pacemaker and was apparently unaware of Cajal’s research, considered these cells to be a real pacemaker of the intestinal muscle layers [5]. The methods needed to prove or disprove these theories were not developed until 7–8 decades later, subsequently proving both to be essentially correct [1].

Between 1925 and 1965, several controversial papers were recorded with scientists presenting different opinions regarding the origin, function and distribution of the ICC. In 1970, after the development of electron microscopy, many mysteries were elucidated. The ICC presents similar properties to smooth muscle cells but also specific characteristics to perform the function of intestinal pacemaker. In the early 1980s, these cells were again brought back to the attention of gastroenterologists by Thunenberg and Faussone-Pellegrini [4,6].

Nowadays, the ICC are the object of study in many medical fields in order to understand the motility of the gut, their involvement in the development of GIST pathology, as well as to elucidate the pathogenesis of various motility disorders [7].

The main function of the digestive system is considered to be the digestion and absorption of nutrients. However, the digestive tract possesses a number of other important functions, namely motor, evacuatory, secretory and incretory immune functions [8]. The motility of the digestive tract is constantly under the control of several regulatory factors, among which are the intrinsic and extrinsic nervous system as well as the humoral system. The main cell types involved in regulating gastrointestinal motor function are enteric neurons, the ICC and effector cells (smooth muscle cells) [9].

The activity of the ICC has been shown to be fundamental, generating and propagating slow electrical waves that regulate the contractile activity of intestinal smooth muscle as well as mediating neurotransmission from intestinal motor neurons towards the intestinal muscle cells. Slow-wave amplitude is under the control of local factors, such as distention and intraluminal chemical stimulation [10,11].

Cajal cell damage leads to hypokinesis and hypotonia of the intestine, which causes severe constipation. In turn, impaired motor function of the digestive tract triggers pathological conditions in the whole body, such as autointoxication due to the absorption of toxic products and deregulation of the absorption of microelements and vitamins, which accelerate the aging process of the body [10].

For this review, we discuss the origin and distribution in the human gastrointestinal tract of the ICC as well as the relationship between these cells and GISTs based on their common ultrastructural features and biomarkers.

## 2. Origin of the Interstitial Cells of Cajal

The gastrointestinal system consists of cells that arise from all embryonic layers [12]. The endoderm gives rise to the epithelial cells of the intestinal lumen and the epithelial cells of the intestinal glands [13]. From the mesoderm, all muscle cells, connective tissue and lymphatic and blood structures are born [12]. Neural crest cells are derived from the ectoderm. Neural crest cells migrate in the digestive system and, after that, they give birth to all enteric neurons [12,13,14].

Despite the fact that the ICC was described more than 100 hundred years ago, their origin has long been an enigma, which is due to the fact that the ICC share structural features with neural crest-derived cells (neurons, glial cells) but also share features with mesoderm-derived cells (muscle cells, fibroblasts) [13]. Despite this debate over the years, some research conducted on chickens, quails [15] and rats [16] demonstrated that the origin of the ICC is in the mesoderm. This fact, demonstrated by the previous research teams, is reinforced by other subsequent studies on rats [2,17].

In the past years, Faussone-Pellegrini [3,6] analyzed the ultrastructural evolution of the ICC in the Auerbach’s plexus and deep muscle plexus. They identified the ICC progenitor cells in newborn rats that are closely related to the nerve fibers of the myenteric plexus, but they could not establish the embryological origin of these cells [12]. Several studies from the past suggested that the ICC is derived from the mesoderm and share the same progenitor cells with cells from the smooth muscle [2,17].

Furthermore, Torihashi studied the evolution of c-kit reactivity markers for smooth muscles and neural crest-derived cell markers on embryonic small intestine specimens from rats [17], thus finding the existence of c-kit+ cells in the outer layers of the intestine in day 12 embryos. However, the cells at this level were not differentiated, and they did not present structural characteristics of an adult muscle cell or the ICC [17].

At embryonic day 15, cells at the level of the circular muscle layer, which are located inside the myenteric plexus, present immunoreactivity for actin and muscle myosin; in contrast, cells that develop at the place where the future longitudinal muscle tissue will be located are c-kit+ cells, but they lack the expression for actin and smooth muscle myosin [17]. In the late stages of embryonic development, a c-kit+ cell subpopulation was found to differentiate into smooth muscle cells showing positive expression for myofilament proteins; consequently, these cells lose the ability to show c-kit+ expression [17]. It is thus assumed that the ICC and smooth muscle cells have a common origin from c-kit+ progenitor cells in the primitive intestine, and all c-kit+ cells will differentiate into the ICC [17].

Another study from the literature carried out in 1998 by Kluppel [2] confirmed the previous results that the ICC and smooth muscle cells have the same embryonic progenitor cell. Kluppel studied the mRNA expression of smooth muscle myosin heavy chain (SMMHC) and c-kit immunoreactivity [2]. He found that all intestinal muscle cells initially show c-kit+ expression and expression for SMMHC; later in the developmental stages, they lose the ability to show c-kit+ expression, while all cells that will further become the ICC will keep c-kit expression but will lose expression for SMMHC [2]. The main conclusion of all this data is that the ICC and smooth muscle cells present the same progenitor cell [12,18].

In conclusion, the origin of the ICC is from the progenitor cells that arise from the mesoderm of the primitive intestine and possess the tyrosine kinase receptor c-kit [19]. However, there are a number of mesenchymal cells with c-kit+ expression that are destined to differentiate into smooth muscle cells, which lose expression for c-kit in the process of development but retain expression for myofilament proteins [18]. Furthermore, information is required to fully understand the entire process [12].

## 3. ICC Distribution in the Human Gastrointestinal Tract

The presence of the ICC in the human gastrointestinal tract has been demonstrated over the years from the esophagus [20] to the anal canal [21]; however, these cells present different morphological features and different tissue distribution.

The ICC exhibit a specific cell position, arrangement and shape based on the localization in different anatomical locations and different layers of the gastrointestinal tract (Figure 1). Therefore, these cells present several cellular subpopulations [22]. In the gastric region, there are reported different subtypes of the ICC and, in the small intestine and colon, the ICC present the same pattern of the subtype of the ICC in each segment [22]. All the subtypes of the ICC have the same ultrastructural characteristics, presence of numerous mitochondria, abundant intermediate filaments and gap junctions with the same cell and smooth muscle cells [22]. The structure of the ICC helps us understand the physiology of the gastrointestinal tract [22].

The ICC include a vast array of specialized cell types within the musculature of the gastrointestinal system. A number of these cell types play a pacemaker role within the gastrointestinal musculature, while others are heavily involved in the modulation of enteric neurotransmission. The most important cell types with a role in intestinal tract motility are the ICC of the myenteric plexus (MP), ICC intramuscular (IM) and ICC of the deep muscle plexus (DMP) (Table 1) [23].

## 4. ICC and GISTs

The concept of “gastrointestinal stromal tumor” was first expressed by Mazur and Clark [28], whom mentioned that all spindle cell tumors from the gastrointestinal system do not present the characteristics of smooth muscle cells. GISTs present diverse structural features, and therefore, a range of tumor subtypes are described, such as plexosarcomas [29] and myenteric plexomas [30]. All of these different features were given only based on the structural features of the progenitor cells. Data from the literature state that tumors with a diameter of less than 3 cm are generally considered benign; however, all GISTs can degenerate malignantly [31]. According to data from the literature, most gastrointestinal tumors are present in the entire gastrointestinal tract with predominant involvement of the stomach (60–70%) but with lesser involvement of the small intestine (20–30%) and large intestine (10%), generally occurring in middle-aged patients [32]. A number of similar tumors have been described with structural features similar to GISTs but with some particular differences and are referred to as extra-gastrointestinal stromal tumors [31].

GISTs often have minimum myoid differentiation and a big number of cytoplasmic filaments [33]. GISTs show ultrastructural features similar to the ICC, namely an elongated cell body and a series of cytoplasmic processes. In order to distinguish ICC from fibroblasts, it should be taken into account that the cytoplasmic processes of fibroblasts are usually thin, very broad, fascia-like structures, whereas the cytoplasmic processes of the ICC are narrow, round in shape and present as slightly flattened. The nucleus is ovoid with one or more nucleoli and has limited content located at the periphery. The cytoplasm of the cell body is in the form of a thin frame around the nucleus often enlarged at the origin of the primary cytoplasmic processes. These extensions are two to five in number, giving rise to numerous secondary and tertiary extensions, and the basal lamina is present but often incomplete [7]. The ultrastructural features between Cajal interstitial cells, smooth muscle cells, fibroblasts and gastrointestinal tumor cells are shown in Table 2.

Diffuse hyperplasia of the ICC is observed in these patients, which is considered a pre-neoplastic lesion [31]. Also in the literature data, it is stated that almost 80% of all GISTs express CD 34 [36]. Germline c-kit mutations have also been detected in patients with GISTs at exons 11 and 13 [37].

GISTs express multiple biomarkers in common with the ICC; some of them, as for example Anoctamin1 and Kit, have been identified as key markers in the diagnosis of GISTs [37].

Additional common biomarkers between the ICC and GISTs have revealed important biological mechanisms in the genesis of GISTs; one of these is ETV1 that is a part of the ETS domain of transcription factors, which holds a key role in the regulation of transcription of ICC and GIST, thereby stimulating tumorigenesis and GIST development [37].

## 5. Discussions

It is assumed that the ICC showing myoid features are not the ICC originally depicted by Raymon y Cajal almost a century ago. The cells around the enteric lymph nodes with immunohistochemical expression of CD117 more likely represent the initially described ICC [7].

The physiological role of these interstitial cells has not yet been fully elucidated. Also, from multiple studies, we can see encouraging information to fully understand the intercellular relationships of the ICC and their role in gastrointestinal tract pathology [38].

The ICC and GISTs show the same precursor cells, most likely stem cells present in the muscle wall of the intestine. These cells retain a certain feature of the evolving germ stem cell and can associate RNA expression of both c-kit and SMMHC protein. Furthermore, the ICC and GISTs share other common biomarkers, such as Anoctamin1 ETV1, which is an important factor in the genesis of GISTs [37] and, in addition, can present muscle cells [7].

New information and data from future studies will be able to clarify all questions regarding this topic.

## Figures and Tables

**Figure 1 medicina-59-00063-f001:**
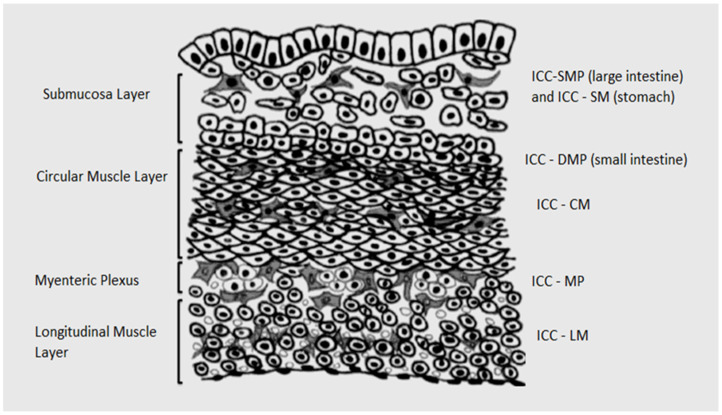
Distribution of the ICC.

**Table 1 medicina-59-00063-t001:** Features of the ICC subtypes.

Subtypes	Distribution	Morphology	Function
ICC progenitor	Stomach Intestine	Groups of round or oval cells	Possess the ability to repair or replenish damaged ICC. [24]
ICC of the myenteric plexus (Auerbach): MY	Stomach Ileon and jejunum Colon	Multipolar cells, possessing multiple interconnected branches	Generate and propagate slow electrical waves. [24]
ICC of longitudinal muscle tissue: LM	Distal oesophagus Stomach Ileon and jejunum Colon	Bipolar cells, oriented along the long axis of the surrounding smoth muscle cells	Important role in neurotransmission between the intestinal nervous system and smooth muscle cells. [25]
ICC of circular muscle tissue: CM	Distal oesophagus Stomach Ileon and jejunum Colon	Bipolar cells, oriented along the long axis of the surrounding smoth muscle cells	Important role in neurotransmission between the intestinal nervous system and smooth muscle cells. [26]
ICC-IM: mixed form, combining ICC-CM and ICC-LM	Distal oesophagus Stomach Ileon and jejunum Colon	Bipolar cells, oriented along the long axis of the surrounding smoth muscle cells	Important role in neurotransmission between the intestinal nervous system and smooth muscle cells. [24]
ICC of deep muscular plexus: DMP	Ileon and jejunum	Multipolar cells, located in the deep muscular plexus; in close association with the nerve fascicles of the deep muscular plexus	Neural transmission. [27]
ICC of submucosa and submucosal plexus: SM and SMP	Pylorus–SM Colon–SMP	Bipolar or Multipolar cells	Pacemakers and neurotransmission roles. [24]

**Table 2 medicina-59-00063-t002:** Cytological characteristics of ICC, Fibroblasts, Smooth muscle cells and GIST.

Cell Type	C-Kit	CV	BL	MIT	NC	IF	RER	GJ
ICC	+	+	-	++	++	++	+	++
Fibroblats	-	-	-	+	+	+	++	+
Smooth muscle cells	+	++	++	++	++	++	+	++
GIST	+	+	-	++	+	+	++	-

Abbreviations: CV = caveolae, BL = basal lamina, MIT = mitochondria, NC = nerve contact, IF = intermediate filaments, RER=Rough endoplasmic reticulum, GJ = gap junction, C = kit reactivity [7,34,35].

## Data Availability

Data sharing is not applicable to this article.

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
