# Peer review of "Interstitial Cells of Cajal—Origin, Distribution and Relationship with Gastrointestinal Tumors"

_medicina, 2022, doi:10.3390/medicina59010063_

Round 1

Reviewer 1 Report

HISTOLOGICAL PICTURES COULD IMPROVE THE OVERALL QUALITY OF THE WORK

Author Response

Hello,

Thank you very much for reviewing the article.

I apologize for not being able to add histological images to this review, due to the fact that we have an ongoing research study program on ICC and the images that have been obtained will be used to publish future research paper . 

Thank you

Sincerely Mihai Zurzu

Reviewer 2 Report

The manuscript entitled as “Interstitial Cells of Cajal – the role in gastrointestinal tumours development” is a mini review that explores the origin, the timeline of their discovery, distribution and their role in gastrointestinal tumors. It is well written and quite pleasant to read. Bellow, the reviewer will point out some topics that could improve the manuscript.

Major observations:

-          No explicit objective is written. I suggest writing down in the introduction (and, therefore, in the abstract) the primary outcome of this review.

-          The reviewer suggests a scheme(s) or an illustration(s), showing the origin, location/distribution, cells around them and summarizing the functions of these cells. Also, a scheme evidencing their similarities and differences with smooth muscle cells, fibroblasts and GIST’s cells would add value to the review.

-          Distribution: Points a) to i) – I suggest a table or a scheme to summarize the information

-          Title – The tumorigenesis of these cells in the present study has less than a page. However, by the title, this part should have the utmost importance. I suggest an alteration in the title like: “Interstitial Cells of Cajal – From its origin/distribution to the role in gastrointestinal tumours development”

-          References: Most of them are quite old.

-          In order to add value to this review, the authors could write another section, as “current status”, focusing on the new discovery from the last 5 years in the field.  

Minor observations:

-          Line 94, change “made” to “performed”.

-          Line 138/139: Gastric region is written two times in the same phrase.

-          Line 145: “inlcud”

Author Response

Hello,

I would like to point out the following changes to the article.

1). I changed the title of the article following your suggestion.

2). Modified the minor spelling and grammatical mistakes in the text.

3). I managed to add to the article a series of illustrations on the distribution of ICCs in the gastrointestinal wall and differences in cell ultrastructure between ICCs, fibroblasts, smooth muscle cell and GIST cells.

4.)  I have taken your advice into account and summarized a number of sentences in the text and of course add some recently dated information on the relationship between these two cellular entities.

5). I know that a lot of the references are old, but I used them because I think that those people (Faussone-Pellegrini, Lecoin, Thuneberg etc.) are pioneers in the area of research of these cells and they deserve to be honoured every time an article is written on this subject.